# The Incidence and Nature of Malpractice Claims against Dentists for Orthodontic Treatment with Periodontal Damage in Israel during the Years 2005–2018—A Descriptive Study

**DOI:** 10.3390/ijerph17238785

**Published:** 2020-11-26

**Authors:** Amir Laviv, Eitan Barnea, Nirit Tagger Green, Rana Kadry, Dima Nassar, Meytal Laviv, Roni Kolerman

**Affiliations:** 1Department of Oral and Maxillofacial Surgery, the Maurice and Gabriela Goldschleger School of Dental Medicine Tel-Aviv University, Tel-Aviv 6997801, Israel; 2Prosthodontist, Private Practice, Tel-Aviv 6100000, Israel; drbarnea@gmail.com; 3Department of Periodontology and Implant Dentistry, the Maurice and Gabriela Goldschleger School of Dental Medicine Tel-Aviv University, Tel-Aviv 6997801, Israel; drnirit@gmail.com (N.T.G.); kolerman@netvision.net.il (R.K.); 4Department of Orthodontics, the Maurice and Gabriela Goldschleger School of Dental Medicine Tel-Aviv University, Tel-Aviv 6997801, Israel; Kadry.rana@yahoo.com; 5Private Practice, Tel-Aviv 6100000, Israel; Dima-11.10@hotmail.com; 6Department of Orthodontics, Hebrew University-Hadassah School of Dental Medicine, Jerusalem 12272, Israel; meytalkiasi@gmail.com

**Keywords:** injury, orthodontic treatment, periodontal treatment, malpractice, lawsuit

## Abstract

In recent years, dental malpractice claims have increased dramatically worldwide. The purpose of the present study is to analyze claims related to orthodontic treatment involving periodontal problems that resulted in legal decisions in Israel. This study analyzed legal claims registered by Medical Consultants International (MCI) between 2005 and 2018. Only closed cases of orthodontic claims involving periodontal problems in which a decision was made were included. The parameters studied included patients’ demographic data, the main reasons of the claim, and complications. Statistical significance was found for aesthetic damage, which was more common in claims of females (*p* = 0.035) and in older claims (*p* = 0.004); tooth damage was more common in claims of older patients (*p* = 0.032); violation of autonomy was higher in private practice (*p* = 0.047) and in more recent claims (*p* = 0.001). As orthodontic treatment is becoming more popular in older patients, and as lawsuit claims become more common in recent years, the orthodontists should always analyze and document the periodontal status of their patients before and during treatment in order to maintain professional practice and avoid future claims.

## 1. Introduction

Legal proceedings against dentists have increased dramatically in recent years and have become a major concern for the dental medicine industry throughout the world [1,2]. The main causes for dental litigation include the combination of private self-payment dental care with high expectations of success, in addition to the increasing number of lawyers willing to take almost all cases, alongside patients who see litigation as a possible solution in any unsuccessful treatment or to their financial problems and personal debts [3,4].

The malpractice claims data are very important for the dentists, as they may advise practitioners to the steps one can make in order to lower the chances of any litigation from occurring [4]. Such claims can be traumatic events for the dentist, and time consuming, as it can mean several weeks out of the dental practice for consultations, preparation, and for a possible trial [5].

The common dental specialty that patients complain about differs from one country to another. However, prosthodontics and oral surgery claims are the major issues in most claims, whereas orthodontic claims are usually less common [2,6].

As orthodontic treatment is usually elective, poorly executed treatment can be frustrating and risk management should be considered. One of the main issues that must be monitored before, during, and after the treatment is the periodontal status of the patient. Patients with periodontal disease, including poor oral hygiene and active periodontal disease, are not good candidates to initiate orthodontic treatment, which may exacerbate and aggravate the periodontal disease during the treatment period [7]. Data regarding dental litigations for orthodontic treatment are very limited, and there are no specific data regarding claims in orthodontic cases specifically involving patients suffering from periodontal diseases.

In Israel, almost 95% of dental practitioners are insured professionally with Medical Consultant International (MCI). Therefore, the data are reliable and available through the insurance company and may well describe the status of orthodontic treatment in Israel.

The aim of the present study is to retrospectively analyze the characteristics of orthodontic treatment involving periodontal problems claims in Israel between 2005 and 2018 based on the computerized database of the MCI insurance company in order to contribute to dental risk management and improve patient safety.

## 2. Materials and Methods

The current study analyzed legal claims registered by MCI from 2005 until 2018, and it was approved by the local international review board (IRB) of Tel-Aviv University.

Inclusion criteria included:(1)All claims related to orthodontic treatment involving periodontal disease.(2)Closed cases in which a decision was determined regarding the claim.(3)Files including full, relevant data: the gender and age of the patient, the date of the complaint, the treatment setting (a private or a public clinic), a detailed description of the adverse event, the type of negligence claimed, and damages awarded for the alleged misconduct.

Exclusion criteria:(1)Open cases which are still in process.(2)Missing relevant data.

All data used by the researchers were anonymous, comprising only filiation data in order to avoid duplication. Collected data included demographic details such as age, sex, the date of the claim, the treatment setting (a private or a public clinic), the complaint and adverse event description, the type of negligence, and damages claimed.

Based on the MCI registry, the collected data were analyzed as follows:The main reasons for the claim—including a lack or delay of diagnosis of periodontal disease, the delay of treatment, a false diagnosis, a change in the treatment plan, and orthodontic treatment on active periodontal disease.Complications or related malpractice—subdivided into distress or pain, violation of autonomy, aesthetic damage, tooth damage or loss, spacing, recession, aggravation of periodontal disease, root resorption, and re-do orthodontic treatment. Each malpractice claim included one or more of the above described alleged damages.Time lag between the beginning of the treatment and the malpractice claim.

### Statistical Analysis

SPSS software (IBM SPSS Statistics for Windows, version 24.0, IBM Corp, Armonk, NY, USA) was used for all statistical analyses. Categorical variables were analyzed using a Chi-squared test or Fisher’s exact test, and continuous variables were analyzed using a *t*-test. A *p*-value of <0.05 was considered statistically significant.

## 3. Results

Between 2005 and 2018, there were more than 1500 claims litigated against dentists regarding all kinds of periodontal malpractice. Out of the closed cases, 35 cases were claims regarding orthodontic treatment combined with periodontal damage preceding or following the orthodontic treatment.

Table 1 describes the general data regarding all closed claim cases. The mean age of the patients was 23.4 ± 10.9 years (range 12–51) whereas 57.1% were older than 19 years. Women comprised 71.4% of the cohort (20/35). Lawsuits against dentists in private practice comprised of 42.9% of cases and the rest against dentists working in a public clinic. Most litigation processes ended in compromise (68.6%) followed by in-court mediation (22.9% of cases). When comparing age and treatment setting, we see that older patients were treated more in private practice compared to younger patients for whom a public clinic was more common. This was close to statistical significance (*p* = 0.062).

The length of orthodontic treatment had been documented in only 25 of the 35 cases (Table 2). Mean treatment time was 3.2 ± 1.2 years, with 64% of the cases treated for 3 years or more.

Table 3 and Table 4 present the reasons of claim and complications, respectively, compared by sex, age (19 years old or less versus older than 19 years old), the treatment setting (private practice or a public clinic), and the year of claim (prior or equal to 2011 versus after 2011). There was no statistical significance in the different reasons of the claim. Orthodontic treatment performed on patients with active periodontal disease was higher in the older age group, which was close to significance (*p* = 0.07). As for the complications, there was a statistical significance regarding aesthetic damage compared by sex (17 claims in females compared to 3 claims in males, *p* = 0.035), violation of autonomy compared by treatment settings (7 cases in private practice compared to 3 cases in public settings, *p* = 0.047), and tooth damage compared by age (16 cases in the older age group compared to 6 cases in the younger age group, *p* = 0.032). There was statistical significance for the recent claims compared to the older claims, in distress and pain and violation of autonomy (more common for recent claims, *p* = 0.01 and *p* = 0.001, respectively), and in esthetic damage and re-do surgery (more common in older claims, *p* = 0.004 and *p* = 0.035, respectively).

Table 5 shows the statistical analysis for treatment length by the reasons for claim. Treatment length was significantly higher in the cases with a delay in diagnosis of periodontal disease and significantly lower in the cases of a lack of diagnosis of periodontal disease.

## 4. Discussion

Dentists are potential targets for compensation lawsuits and sometimes face unnecessary risks of legal action concerning their treatment [6]. Treating physicians should always bear in mind simple risk management strategies for the dual purposes of rendering an enhanced level of treatment and minimizing exposure to potential legal action [7]. For orthodontic treatment, such strategies include a prerequisite in any patient seeking orthodontic treatment to achieve periodontal health and, therefore, a periodontal diagnosis including oral examination, periodontal charting, and a complete periapical radiographic series should always be carried out before the beginning of the orthodontic therapy [6,7,8]. Moreover, comprehensive records should be taken before, during, and after treatment as well as obtaining informed consent from each patient thus discussing the limitations of treatment [6,7,8]. The main reasons for lawsuits in orthodontics include the more costly procedures, the generally longer treatment time, which usually involve aesthetics, and treatment performed by dentists working without adequate training [9]. When properly used, orthodontic treatment can improve tooth positions, creating access for oral hygiene, and altering occlusal factors [10]. On the other hand, it can lead to additional attachment loss due to plaque accumulation and gingival inflammation in those patients with previous periodontal disease. Strict biofilm control and periodontal 1–3-month maintenance programs are essential in the active phase of orthodontic treatment [11]. Furthermore, orthodontic forces must be carefully applied in teeth with a reduced periodontium [12].

Younger patients generally have a healthy periodontium [13], while older patients may have higher odds of an underlying periodontal disease, which could worsen during orthodontic therapy. The effect of orthodontic treatment upon periodontal tissues and the risks and benefits of orthodontic tooth movement in patients with periodontal pathology is controversial [14]. Therefore, as mentioned, it is important for orthodontists to identify periodontal disease before orthodontic treatment and sequence the orthodontic and periodontal therapy correctly [14,15]. The current cohort demonstrates a few interesting findings for the orthodontic patients with periodontal disease.

There was no difference in the number of claims in private practice (43% of claims) compared to public clinic (57% of claims), as seen in a previous article in which medical accidents did not significantly differ according to the facility type comparing hospital-based practices and private practices [8].

In the last decade, orthodontic patients’ age trend has changed, from children and adolescents to an increasing number of adult patients [15]. These findings correlate to the age of patients’ claims in the current study, with 57% of claimers older than 19 years compared to 25 years (47.3% of patients) in a previous study [8].

The present study demonstrates that the most the common cause for claims was related to either orthodontic treatment pursued on active periodontal disease (32 cases) or late diagnosis and delay of treatment of an active periodontal disease (34 cases), but without statistical significance in the different sex, age, treatment setting, or the year of claim. Complications caused by the treatment included distress and pain (43%), violation of autonomy (29%), aesthetic damage (57%), tooth damage (63%), spacing (40%), root recession (29%), periodontal disease or aggravation (97%), re-do surgery (11%), root resorption (6%), TMJ injury (3%), and re-do orthodontic treatment (6%).

In Israel, as in most countries, signing informed consent is obligatory before performing clinical treatment. Waving this action is considered violation of patients’ autonomy, and is a common reason for claim [2,16]. In our cohort, violation of autonomy was more common in private practice compared to public clinics (*p* = 0.047) and more common later than 2011 compared to treatment before 2011 (*p* = 0.001). This can be explained by the strict protocol in public clinics, where each patient must sign different documents before treatment, including a financial agreement and an informed consent form. Additionally, public corporations have more administrative staff to regulate the collection of patients’ signed informed consent forms. Forgetting an informed consent form signing may happen, which may be interpreted as violation of autonomy later on, especially if there are problems arising throughout the treatment.

As our cohort deals with orthodontic treatment, including periodontal complications, the older the patient the probability for periodontal involvement prior to or during treatment is higher and tooth loss can be more frequent in these patient groups. Aesthetic damage was more common in females, which is not surprising as females have more aesthetic demands, greater interest in dental health, and they usually use the services more than men [3,9,17]. This is also in line with reports in orthodontic treatment claims in which the main subject of dispute was dissatisfaction with appearance, a more common claim for females [8].

As for the treatment length for the different claim reasons, treatment length was higher for patients with delay in diagnosis of periodontal disease (*p* = 0.034) and lower for patients with lack of diagnosis of periodontal disease (*p* = 0.044). These findings can be explained by the probability of postponing the treatment once periodontal disease is detected in order to control the periodontal disease.

### Strengths and Limitations

The MCI database used in this study covers the entire country, as almost 95% of the dental practitioners in Israel were professionally insured by this company during the 14-year study period. This is a major strength of the present study.

However, the study also has several limitations. First, the relevance of the results for the subgroups divided by age, sex, and the treatment setting (a private or public clinic) is limited, as the division of the total number of patients undergoing orthodontic treatment according to these subgroups is unknown. For example, although the majority of claimants were female, it is not possible to conclude that it is therefore riskier to treat a woman from a malpractice perspective; instead, the higher proportion of claims from women may simply reflect the fact that women represent the majority of treated patients. Second, although the total number of claims filed during the years examined in the study were described, only settled cases were analyzed. Third, data on the compensation payments were not included as the insurance company objected to this.

## 5. Conclusions

The main errors involved in orthodontic treatment claims include treatments below the standard of care and given dissatisfaction with the treatment outcome. As orthodontic treatment becomes more common in older patients, and as lawsuit claims become more common in recent years, crucial steps for treatment should always be implemented. These include taking comprehensive records before, during, and after treatment, clearing patients for dental problems (restorative, prosthodontic, periodontal), discussing the treatment plan in detail with an explanation of all the benefits and complications of treatment, and obtaining informed consent. This routine behavior will maintain professional practice and avoid future claims.

## Figures and Tables

**Table 1 ijerph-17-08785-t001:** Data of compensated injury.

Variables	*n*	(%)
Age		
≤19	15	−42.9
>19	20	−57.1
Gender		
Male	10	−28.6
Female	25	−71.4
Sector		
Private practice	15	−42.9
Public clinic	20	−57.1
Litigation status		
Compromise	24	−68.6
Court mediation	8	−22.9
Rejection	2	−5.7
Closed not covered	1	−2.9
Total	35	−100

**Table 2 ijerph-17-08785-t002:** Length of orthodontic treatment as documented in 25 cases.

Treatment Length (Years)	*n*	%
1.5	1	(4)
2.0	6	(24)
2.5	1	(4)
2.8	1	(4)
3.0	7	(28)
4.0	4	(16)
4.5	1	(4)
4.8	1	(4)
5.0	2	(8)
6.0	1	(4)
Total	25	100%

**Table 3 ijerph-17-08785-t003:** Reasons of claim divided by gender, age, and treatment settings.

Reason for Claim (*n*, %)	Sex (*n*)			Age (*n*)			Treatment Setting (*n*)			Year of Claim (*n*)		
	F (25)	M (10)	*p*-Value	<19 (15)	>19 (20)	*p*-Value	Private (15)	Public (20)	*p*-Value			
Delay of diagnosis (15, 42.9%)	9	6	0.179	6	9	0.521	7	8	0.479	7	8	0.596
Delay of treatment (34, 97.1%)	24	10	0.714	14	20	0.429	14	20	0.429	15	19	0.547
False diagnosis (5, 14.3%)	5	0	0.164	2	3	0.640	1	4	0.272	3	2	0.415
Treatment plan change (1, 2.9%)	1	0	0.714	0	1	0.571	1	0	0.429	0	1	0.543
Lack of diagnosis of periodontal disease (19, 54.3%)	15	4	0.243	8	11	0.596	7	12	0.330	8	11	0.449
Orthodontic treatment on active periodontal disease (32, 91.4%)	23	9	0.649	12	20	0.070	14	18	0.610	13	19	0.086

**Table 4 ijerph-17-08785-t004:** Complications divided by gender, age, and treatment settings.

Complications (*n*, %)	Sex (*n*)			Age (*n*)			Treatment Setting (*n*)			Year of Claim (*n*)		
	F (25)	M (10)	*p*-Value	<19 (15)	>19 (20)	*p*-Value	Private (15)	Public (20)	*p*-Value	≤2011 (16)	>2011 (19)	*p*-Value
Distress and pain (15, 42.9%)	15	5	0.433	4	11	0.091	9	6	0.076	3	12	0.010
Violation of autonomy (10, 28.6%)	7	3	0.606	3	7	0.279	7	3	0.047	0	10	0.001
Esthetic damage (20, 57.1%)	17	3	0.035	9	11	0.590	9	11	0.427	13	7	0.004
Tooth damage/loss (22, 62.9%)	15	7	0.440	6	16	0.032	11	11	0.226	9	13	0.347
Spacing (14, 40%)	12	2	0.125	5	9	0.365	6	8	0.635	8	6	0.223
Root recession (10, 28.6%)	9	1	0.129	5	5	0.433	5	5	0.433	6	4	0.243
Periodontal disease or aggravation (34, 97.1%)	25	9	0.286	15	19	0.571	14	20	0.429	16	18	0.543
Re-do surgery (4, 11.4%)	2	2	0.319	3	1	0.200	1	3	0.419	4	0	0.035
Root resorption (2, 5.7%)	1	1	0.496	2	0	0.176	1	1	0.681	2	0	0.202
Temporomandibular joint (TMJ) injury (1, 2.9%)	1	0	0.714	1	0	0.429	1	0	0.429	1	0	0.547
Re-do ortho treatment (2, 5.7%)	2	0	0.504	1	1	0.681	1	1	0.681	2	0	0.202

**Table 5 ijerph-17-08785-t005:** Distribution of treatment length in the different reasons for claim.

Reason for Claim (*n*, %)	Mean of Treatment Length	*p*-Value
Delay of diagnosis (11, 44%)	3.8 ± 1.1489	0.034
Delay of treatment (24, 96%)	3.254 ± 1.2065	0.838
False diagnosis (4, 16%)	3.250 ± 0.9574	0.991
Lack of diagnosis of periodontal disease (13, 52%)	2.792 ± 1.0889	0.044
Ortho treatment on active periodontal disease (22, 88%)	3.105 ± 1.1454	0.112

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
