# Peer review of "The Incidence and Nature of Malpractice Claims against Dentists for Orthodontic Treatment with Periodontal Damage in Israel during the Years 2005–2018—A Descriptive Study"

_ijerph, 2020, doi:10.3390/ijerph17238785_

Round 1
Reviewer 1 Report
I am impressed with your research.
I think it would be interesting to investigate the detection of periodontal disease before and during orthodontic treatment.
Only a minor suggestion, why don't you check how many orthodontists use phase-contrast microscopy in order to check sub-gingival plaque and what about the oral hygiene protocols proposed in adult or older patients?
Author Response
Dear reviewer,
thank you very much for your thorough review.
regarding the suggestion - unfortunately, as the research was performed using MCI data only (without opening patient's files), we could not get this information. However, this is a great suggestion for future research, that we would definitely take into consideration.
Thank you very much.
Amir Laviv
Reviewer 2 Report
The Section Materials and Methods must be improved

Author Response
Dear reviewer,
Thank you for the thorough review. We take this opportunity to thank you for the most constructive commentaries that largely helped us to ameliorate our work. We have made an effort to address all the reviewers’ comments.
Please find the revised manuscript.
All comments addressed as answers as follows:
Line 2 - Please clarify the type of the article, eg Research?
Type of research added to the title.
Line 69 - Reference(s)? What about exclusion criteria?
Changed in the materials and methods section
Line 73 - Replace by ''gender''...
Fixed in the article
Line 76 - Reference(s)?
This is the innovation of the study. Therefore, no references added.
Line 86 - I think that a regression analysis model would be more appropriate. Chi-sq. test is weak and unreliable. Moreover, a regression model would control possible confounders...
The cohort in the study is all MCI lawsuits in orthodontic treatment with periodontal problems, with no control group (descriptive study of this population). Therefore, regression model for lawsuit (yes or no – lawsuit) can not be performed. This is the reason for choosing chi-sqare analysis.
Line 87 - What about the normality of the distribution?
Continuous variables were analyzed for normal distribution using Kolmogorov-Smirnov test as well as Q-Q plots with normality tests.
Line 160 - In which study? The current? If so, please do not repeat Results....
Changed in the text, as these results are important findings to discuss.
Line 171 - References?
Elaborated in the text
Line 177 - Do not repeat Results....
Fixed in the text, only discussion of the issue left
Line 183 - Do not repeat Results and state Reference(s) for possible conclusions...
The findings are important, and possible explanation appear later.
Line 188 - Possible limitations?
Added to the text
Line 190 - We do not state Reference(s) in this Section.... Moreover, reduce it and state the main Conclusion(s) only....
Fixed in the text
We hope that the revised manuscript in its present form will be suitable for publication in the international journal of Environmental Research and Public Health. We look forward to hearing from you.
Sincerely,
Amir Laviv
